# Temporal Saliency-Guided Distillation: A Scalable Framework for Distilling Video Datasets

## Abstract

Dataset distillation (DD) has emerged as a powerful paradigm for dataset compression, enabling the synthesis of compact surrogate datasets that approximate the training utility of large-scale ones. While significant progress has been achieved in distilling image datasets, extending DD to the video domain remains challenging due to the high dimensionality and temporal complexity inherent in video data. Existing video distillation (VD) methods often suffer from excessive computational costs and struggle to preserve temporal dynamics, as naïve extensions of image-based approaches typically lead to degraded performance. In this paper, we propose a novel uni-level video dataset distillation framework that directly optimizes synthetic videos with respect to a pre-trained model. To address temporal redundancy and enhance motion preservation, we introduce a temporal saliency-guided filtering mechanism that leverages inter-frame differences to guide the distillation process, encouraging the retention of informative temporal cues while suppressing frame-level redundancy. Extensive experiments on standard video benchmarks demonstrate that our method achieves state-of-the-art performance, bridging the gap between real and distilled video data and offering a scalable solution for video dataset compression.

## 1 Introduction

In recent years, video data has accounted for over 70% of global internet traffic, and this proportion continues to grow steadily Barnett et al. (2018). The explosive increase in online video content has driven a strong demand for automatic video understanding, which has in turn led to remarkable progress in video action recognition using neural networks. However, the impressive performance of these models often comes at the cost of high computational expense (Wang et al., 2021). The sheer volume of video data, along with its significantly larger size compared to images, imposes substantial training costs and computational burdens on the models. Dataset distillation has become a promising technique to address this challenge, aiming to compress large-scale datasets into compact synthetic datasets while preserving as much task-specific information as possible.

However, existing dataset distillation methods still primarily focus on the image domain and are difficult to directly extend to videos. Existing methods can be broadly classified into

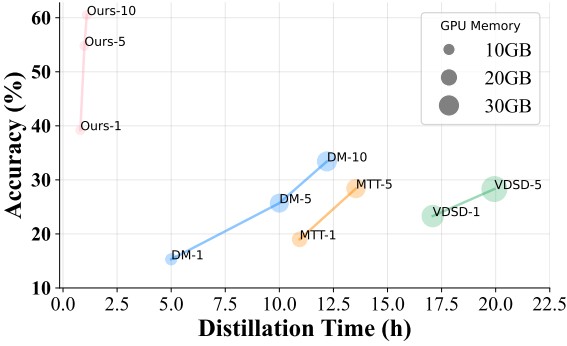

Figure 1: Comparison of performance and computational cost between different methods across all the IPC settings. The size of each point indicates memory consumption, and the labels denote the corresponding method names and IPC (instance per class) settings. The methods involved include DM (Zhao et al., 2021), MTT (Cazenavette et al., 2022), VDSD (Wang et al., 2024) and Ours. IPC represents the number of instances per class in the distilled dataset and serves as a primary metric for indicating the compression rate. As shown in the figure, Our method exhibits superior performance.

two categories: optimization-based and training-free methods. Traditional optimization-based methods (Wang et al., 2018; Zhao et al., 2021; Nguyen et al., 2021; Zhao & Bilen, 2021; Cazenavette et al., 2022; Zhong et al., 2024a; Sajedi et al., 2023; Cazenavette et al., 2023; Zhong et al., 2024b) typically employ a bi-level optimization framework, wherein the synthetic dataset is optimized in conjunction with the training of a neural network that functions as a task-specific information extractor. Another category of optimization-based methods, decoupled distillation methods (Yin et al., 2024; Shao et al., 2024a; Yin & Shen, 2024; Du et al., 2024; Shao et al., 2024b), optimize synthetic datasets without repeatedly training random sampled neural networks to further reduce cost. Although these methods often achieve strong performance under high compression ratios, the need to optimize all the pixels with a synthetic dataset introduces considerable time overhead and memory consumption. In contrast, training-free methods (Sun et al., 2024; Su et al., 2024; Gu et al., 2024; Chen et al., 2025) commonly utilize the priors from pre-trained classifiers or generative models to directly synthesize datasets, avoiding pixel-level optimization of individual images and thereby significantly reducing computational costs. However, due to their limited capacity to remove redundant information, such methods generally result in suboptimal performance.

Although effective for image datasets, these two types of methods encounter significant challenges when applied to video datasets. Videos comprise sequences of frames, which significantly amplifies the computational cost of traditional optimization-based approaches, making them impractical for high-resolution or large-scale video distillation scenarios. Moreover, the temporal dimension inherent in video data necessitates that the distillation process account for inter-frame relationships to preserve temporal coherence and motion dynamics. In contrast, training-free methods are limited by the current capabilities of video generation models. While image generation techniques have reached a level of maturity that allows for the synthesis of high-quality images, video generation models still struggle to produce high-fidelity sequences suitable for action recognition tasks, often suffering from fragmented or incoherent temporal dynamics. Additionally, the presence of temporal dynamics greatly increases the difficulty of selecting informative frames, further contributing to the suboptimal performance of these approaches.

To address these challenges, VDSD (Wang et al., 2024) introduces a two-stage video distillation paradigm. In the first stage, it randomly selects frames from each video and performs standard image-level distillation. In the second stage, an interpolation model is used to reconstruct videos, followed by image distillation applied to each individual frame. Although this method achieves certain performance, the two stages are inherently conflicting and fail to effectively incorporate temporal information. As an extention of VDSD, IDTD (Zhao et al., 2024) selects frames with maximal differences to enhance frame diversity in the first stage and randomly sampling frames for interpolation in the second stage, thereby increasing video variability. While IDTD demonstrates improvements over VDSD, it maintains the same distillation pipeline and still overlooks essential temporal dependencies. Furthermore, the two-stage framework introduces considerable time overhead while offering only marginal performance gains.

However, extending image dataset distillation methods to videos and preserving temporal dynamics during the distillation process is a non-trivial task. First, modeling temporal dynamics is itself a difficult task. While using optical flow as a representation is a straightforward solution, it introduces considerable storage and computational overhead, necessitating more compact and efficient alternatives. Second, performing data augmentation on video without disrupting its temporal coherence is another major concern. Differentiable data augmentation has proven effective in enhancing the diversity of synthetic data in dataset distillation. However, naively applying image-level augmentation techniques to videos can degrade them into a sequence of unrelated frames, ultimately impairing performance. Therefore, designing augmentation strategies that respect the temporal structure of videos is crucial for effective video dataset distillation.

In this paper, we propose a novel framework for video dataset distillation. By simplifying the conventional bi-level optimization framework, we design an efficient uni-level optimization-based distillation process. Furthermore, we introduce the **T**emporal **S**aliency-**G**uided **F**ilter (TSGF) to preserve and enhance temporal information during the distillation process. Specifically, we first train a standard video classification model to encode video information into a pre-trained classifier. Subsequently, we align the distribution of the synthetic dataset with the model's internal statistics under the guidance of classification loss. To further reinforce temporal information, we apply TSGF, computed via inter-frame differences, as a constraint during optimization. In the post-evaluation phase, we propose a dynamic video augmentation strategy that enhances synthetic videos with

TSGF to identify the key frames. By effectively capturing and reinforcing temporal characteristics throughout the overall distillation process, our method achieves significant performance improvements across various benchmark datasets. As illustrated in Figure 1, compared to existing image and video distillation methods, our approach significantly improves performance while substantially reducing both computational time and memory consumption, demonstrating the effectiveness of our framework.

The contributions of this paper can be summarized as follows:

- We propose an efficient uni-level video dataset distillation framework that is orthogonal to existing methods. It effectively addresses the excessive time consumption of prior approaches and reduces framework complexity.
- We introduce TSGF, a temporal saliency-guided filter to guide the distillation process, along with a dynamic data augmentation technique. Both components enhance the temporal consistency and informativeness of the generated videos.
- Extensive experiments across various video datasets and compression ratios demonstrate the effectiveness and efficiency of our method, significantly reducing training time for video action classification tasks.

## 2 RELATED WORK

### 2.1 IMAGE DATASET DISTILLATION

Traditional optimization-based dataset distillation methods were initially formulated as meta-learning problems (Wang et al., 2018), in which the objective is achieved by alternately updating the neural network and the distilled dataset. To mitigate the substantial computational cost caused by the unrolled computational graph inherent in meta-learning, DC (Zhao et al., 2021) proposed matching the gradients of the loss function with respect to model parameters on both real and synthetic datasets, thereby significantly reducing overhead via short-horizon gradient matching. MTT (Cazenavette et al., 2022) extended this idea by aligning the parameter trajectories of models trained on real and synthetic datasets, enabling long-horizon matching. Alternatively, DM (Zhao et al., 2021) focused on aligning feature distributions between datasets by treating the neural network solely as a feature extractor, further reducing computational costs.

In contrast to optimization-based methods, training-free dataset distillation approaches avoid pixel-level optimization of the synthetic dataset and instead aim to preserve high-level semantic information. RDED (Sun et al., 2024) and DDPS (Zhong et al., 2024c) respectively leverage a pre-trained classifier and a diffusion model to directly select the most class-representative images from the original dataset. Other methods such as Minimax (Gu et al., 2024), D$^4$M (Su et al., 2024), and IGD (Chen et al., 2025) fine-tune components of diffusion models (Song et al., 2021) to directly generate synthetic images for the target dataset. While these approaches demonstrate promising performance on large-scale datasets (Deng et al., 2009), they tend to suffer under extreme compression ratios due to a lack of optimization, often resulting in synthetic datasets with redundant information and suboptimal performance.

Distinct from the aforementioned methods, decoupled dataset distillation methods (Yin et al., 2024; Yin & Shen, 2024; Shao et al., 2024b;a; Du et al., 2024), such as SRe$^2$L (Yin et al., 2024), first compress the information of the full dataset into a target model, and then generate synthetic datasets by aligning their statistical properties (e.g., batch normalization statistics) and task-specific loss with those of the model. However, a key limitation of existing work (Chen et al., 2024) that directly applies decoupled methods to video datasets lies in the treatment of frames from the same video as independent and unrelated samples, resulting in redundant optimization and neglect of temporal structure.

### 2.2 VIDEO DATASET DISTILLATION

Compared to image data, video data incorporates an additional temporal dimension, which significantly increases the complexity of the distillation process. Currently, research on video dataset distillation remains in its early stages. VDSD (Wang et al., 2024) is the first work in this field, introducing a two-stage paradigm that disentangles the static and dynamic components of video

---

**Algorithm 1** Saliency-Guided Video Distillation

---

**Input:** Original dataset $\mathcal{T}$, distillation iterations $K$, learning rate $\eta$
  Train $\theta_{\mathcal{T}}$ on $\mathcal{T}$: $\theta_{\mathcal{T}} = \arg\min_{\theta_{\mathcal{T}}} \mathcal{L}_{ce}(\phi_{\theta_{\mathcal{T}}}(\mathbf{x}_T), y_T)$

  **for** $k \leftarrow 0$ to $K-1$ **do**
    Calculate regularization loss $\mathcal{L}_{reg}$ using Eq. (5)
    Calculate distillation loss $\mathcal{L} = \mathcal{L}_{ce}(\phi_{\theta_{\mathcal{T}}}(\mathbf{x}_S), y) + \mathcal{L}_{reg}$
    Compute mask $M$ using Eq. (8), where compute temporal saliency vector $s$ using Eq. (6) and Eq. (7)
    Update $\mathcal{S} \leftarrow \mathcal{S} - \eta M \nabla_{\mathcal{S}} \mathcal{L}$
  **end for**
  Compute temporal saliency vector $s$ using Eq. (6) and Eq. (7)
  Apply temporally guided video augmentation: $\mathbf{x}_S = D(\mathbf{x}_S)$
  Recalibrate video data labels: $y_S = \phi_{\theta_{\mathcal{T}}}(\mathbf{x}_S)$
**Output:** Distilled dataset $\mathcal{S} = (\mathbf{x}_{\mathcal{S}}, y_{\mathcal{S}})$

---

data. In the static distillation stage, VDSD compresses pixel-level information into a single frame sampled from the video, aiming to align spatial features between the original and synthetic data. In the dynamic distillation stage, the distilled image is interpolated into a video, and semantic motion information is utilized to achieve temporal alignment between the original and synthetic sequences.

IDTD (Zhao et al., 2024) builds upon this two-stage paradigm and achieves further progress by incorporating two key modules. The Information Diversification module augments the distilled data into multiple feature segments to enhance diversity, while the Temporal Densification module aggregates these segments into complete video clips to capture richer temporal dynamics. These components improve the algorithm's capacity to compress temporal features, leading to better performance.

Despite these advances, existing video dataset distillation methods often suffer from conflicting optimization objectives and considerable computational overhead. In contrast to previous approaches, we utilize a pre-trained classifier to guide the generation of synthetic video datasets, effectively reducing computational cost while preserving temporal coherence to the greatest extent possible.

## 3 METHOD

### 3.1 PRELIMINARY

The objective of dataset distillation is to compress a large-scale training set $\mathcal{T} = \{(\mathbf{x}_{\mathcal{T}}^i, y_{\mathcal{T}}^i)\}_{i=1}^{|\mathcal{T}|}$ into a distilled dataset $\mathcal{S} = \{(\mathbf{x}_{\mathcal{S}}^i, y_{\mathcal{S}}^i)\}_{i=1}^{|\mathcal{S}|}$ ($|\mathcal{S}| << |\mathcal{T}|$), while preserving the training accuracy as much as possible. The learning objective on $\mathcal{S}$ can be formulated as follow:

$$\theta_{\mathcal{S}} = \arg\min_{\theta} \mathbb{E}_{(\mathbf{x}_{\mathcal{S}}, y_{\mathcal{S}}) \in \mathcal{S}}[l(\phi_{\theta_{\mathcal{S}}}(\mathbf{x}_{\mathcal{S}}), y_{\mathcal{S}})], \tag{1}$$

where $l(\cdot, \cdot)$ denotes the typical loss function (e.g., cross-entropy loss), and $\phi_{\theta_{\mathcal{S}}}$ represents the neural network with parameter $\theta_{\mathcal{S}}$. The primary objective of the dataset distillation task is to generate synthetic data aimed at attaining a specific or minimal performance disparity on the original validation data when the same models are trained on the synthetic data and the original dataset, respectively. Thus, we aim to optimize $\mathcal{S}$ as follow:

$$\arg\min_{\mathcal{S}, |\mathcal{S}|}(\sup\{|l(\phi_{\theta_{\mathcal{T}}}(\mathbf{x}_{val}), y_{val}) - l(\phi_{\theta_{\mathcal{S}}}(\mathbf{x}_{val}), y_{val})|\}_{(\mathbf{x}_{val}, y_{val}) \in \mathcal{T}}), \tag{2}$$

where $(\mathbf{x}_{val}, y_{val})$ denotes the image and label pair sampled from the validation set of $\mathcal{T}$. For a typical dataset distillation process, the synthetic dataset $\mathcal{S}$ is first initialized using random noise or randomly sampled instances. Then, the distillation loss is computed, and $\mathcal{S}$ is optimized based on the loss. However, for video datasets, each video consists of multiple frames and can be regarded as image data with a large batch size in terms of computational cost. Consequently, previous image-based distillation methods are difficult to apply due to the substantial overhead of video datasets.

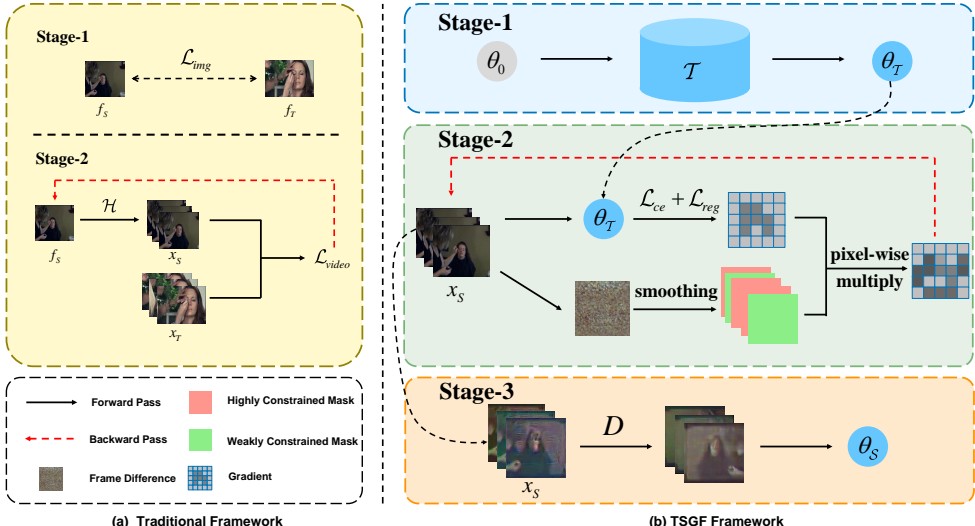

Figure 2: Comparison between our TSGF framework and the traditional two-stage video distillation paradigm. $\mathcal{S}$ and $\mathcal{T}$ represent the synthetic dataset and the real train dataset, respectively. $f$ represents an individual frame and $x$ denotes a video. $\theta$ is the model parameters. While both stages of the traditional approach primarily rely on pixel-level information, our unified framework effectively distills temporal information through a three-stage process. TSGF comprises two key components: $\text{TSGF}_O$ and $\text{TSGF}_A$. During the optimization stage, $\text{TSGF}_O$ constrains the optimization by computing inter-frame differences. In the evaluation stage, $\text{TSGF}_A$ guides the data augmentation process to preserve temporal dynamics.

## 3.2 UNI-LEVEL VD FRAMEWORK

Distinct from existing two-stage video distillation frameworks, we propose a novel uni-level video dataset distillation paradigm that enables efficient synthetic data generation. By decoupling model training from synthetic data optimization, our approach effectively addresses the substantial intra-batch computational cost encountered by traditional image-based distillation methods when applied to video data.

To capture the dataset distribution, we first train a video action recognition model, compressing the informative content of the original dataset into the model. The training process is formulated as:

$$\theta_{\mathcal{T}} = \arg \min_{\theta_{\mathcal{T}}} \mathcal{L}_{ce}(\phi_{\theta_{\mathcal{T}}}(\mathbf{x}_{\mathcal{T}}), y_{\mathcal{T}}), \tag{3}$$

where $\theta_{\mathcal{T}}$ denotes the parameters of the pre-trained model, and $\mathcal{L}_{ce}$ is the cross-entropy loss. Subsequently, we optimize the synthetic video samples using class-discriminative and statistical information extracted from the pre-trained model, following a process similar to (Yin et al., 2020):

$$\mathbf{x}_{\mathcal{S}} = \arg \min_{\mathbf{x}_{\mathcal{S}}} \alpha_{ce} \mathcal{L}_{ce}(\phi_{\theta_{\mathcal{T}}}(\mathbf{x}_{\mathcal{S}}), y) + \alpha_{reg} \mathcal{L}_{reg} \tag{4}$$

where $y$ is the one-hot class label assigned to $\mathbf{x}_{\mathcal{S}}$, $\mathcal{L}_{reg}$ is a regularization term used to align the statistical distribution of synthetic data with the pre-trained model, and $\alpha_{ce}$ and $\alpha_{reg}$ are coefficients used for weighting. The regularization loss is computed as follows:

$$\mathcal{L}_{reg} = \sum_l \left( ||\mu_l(\mathbf{x}_{\mathcal{S}}) - RM_l||_2 + ||\sigma_l^2(\mathbf{x}_{\mathcal{S}}) - RV_l||_2 \right) \tag{5}$$

where $l$ denotes the index of the batch normalization (BN) layer, $\mu_l(\mathbf{x}_{\mathcal{S}})$ and $\sigma_l^2(\mathbf{x}_{\mathcal{S}})$ represent the mean and variance of the activations in the $l$-th BN layer, and $RM_l$, $RV_l$ are the corresponding running mean and running variance.

### 3.3 TEMPORAL SALIENCY-GUIDED FILTER

Naively treating video frames as discrete and independent images and applying uniform optimization across the entire video often leads to the loss of motion-related information in the temporal dimension, resulting in suboptimal performance (Chen et al., 2024). To address this limitation, we propose a **T**emporal **S**aliency-**G**uided **F**ilter (TSGF), which comprises temporally guided video optimization and temporally guided video augmentation. This filter guides the update of video data by preserving and enhancing temporal coherence, thereby improving the overall quality of the distilled videos.

**Temporally Guided Video Optimization.** To preserve motion-related semantic information during the optimization process, it is essential to impose constraints that prevent over-optimization, which could otherwise diminish temporal dynamics. To this end, we compute the temporal saliency of each frame and assign adaptive optimization magnitudes accordingly. Specifically, we calculate the inter-frame difference for each frame using the following formulation:

$$d_i = \frac{|f_{i+1} - f_i| + |f_i - f_{i-1}|}{2} \tag{6}$$

where $f_i$ denotes the $i$-th frame, and $d_i$ represents the inter-frame difference of the $i$-th frame. Inter-frame differences are often used as a metric to determine key frames in a video, and the magnitude of these differences can indicate the importance of each frame to some extent. However, the raw inter-frame difference cannot be directly used to measure the temporal saliency of the video frames because of the local jitter. To take advantage of the locality inherent in the video and eliminate the effects of local jitter, we apply smoothing to the raw inter-frame differences as follows:

$$s_i = \alpha_0 * d_i + \alpha_1 * d_{i-1} + \cdots + \alpha_k * d_{i-k}, \tag{7}$$

where $s_i$ denotes the temporal saliency of the $i$-th frame, $k$ represents the window length, and $\alpha_k$ is the smoothing weight for previous frames, which is determined by the window function. To preserve the temporal information of the video, we compute the TSGF based on $s_i$ of each frame. Specifically, the greater the importance of a frame, the smaller its optimization degree will be. The calculation formula is as follows:

$$M = \frac{\max(\epsilon - s, 0)}{\max(s) - \min(s)}, \tag{8}$$

where $M$ denotes the optimization mask, $s$ represents the temporal saliency vector of the video, and $\epsilon$ indicates the upper bound of temporal saliency. The resulting mask is directly multiplied with the gradients to guide the optimization process for each frame in the video.

**Temporally Guided Video Augmentation.** To further enhance the diversity of synthetic datasets while preserving teporal information, we apply a dynamic video data augmentation during post-evaluation phase. To prevent the loss of key frame information during augmentation, we also utilize TSGF as guidance to selectively perform data augmentation. The augmented videos are then used for the final model training.

Common image augmentation methods such as MixUp (Zhang et al., 2017) and CutMix (Yun et al., 2019) are not directly applicable to videos, as they can disrupt the inherent temporal structure. To address this issue, we first compute $s_i$ of each video frame using the Eq. (7). Then, we apply VideoMix (Yun et al., 2020) augmentation at the same positions across non-key frames, guided by their temporal saliency scores $s_i$. Specifically, when $s_i \leq \epsilon$, data augmentation is applied to the frame $f_i$, and if not, the frame remains unchanged. This strategy is simple yet effective, and the detailed ablation results can be found in the Appendix C. Our distillation process is summarized in Algorithm 1.

## 4 EXPERIMENTS

To verify the efficiency of our proposed method, we conduct experiments following two-steps paradigm, as the standard evaluation procedure for dataset distillation. In the first step, we train a randomly initialized model using the original dataset to compress the information of the original dataset into the model, and then use the model to synthesize the distilled dataset. In the second step, a standard model training process is conducted using the synthetic dataset, and performance is evaluated on the original test set.

Table 1: Comparison of top-1 accuracy with existing methods on small-scale datasets. Our method achieve significant performance improvements across various settings.

| Dataset | | MiniUCF | | HMDB51 | |
|---|---|---|---|---|---|
| IPC | | 1 | 5 | 1 | 5 |
| Coreset Selection | Random | $9.9 \pm 0.8$ | $22.9 \pm 1.1$ | $4.6 \pm 0.5$ | $6.6 \pm 0.7$ |
| | Herding | $12.7 \pm 1.6$ | $25.8 \pm 0.3$ | $3.8 \pm 0.2$ | $8.5 \pm 0.4$ |
| | K-Center | $11.5 \pm 0.7$ | $23.0 \pm 1.3$ | $3.1 \pm 0.1$ | $5.2 \pm 0.3$ |
| Dataset Distillation | DM | $15.3 \pm 1.1$ | $25.7 \pm 0.2$ | $6.1 \pm 0.2$ | $8.0 \pm 0.2$ |
| | MTT | $19.0 \pm 0.1$ | $28.4 \pm 0.7$ | $6.6 \pm 0.5$ | $8.4 \pm 0.6$ |
| | FRePo | $20.3 \pm 0.5$ | $30.2 \pm 1.7$ | $7.2 \pm 0.8$ | $9.6 \pm 0.7$ |
| | DM+VDSD | $17.5 \pm 0.1$ | $27.2 \pm 0.4$ | $6.0 \pm 0.9$ | $8.2 \pm 0.4$ |
| | MTT+VDSD | $23.3 \pm 0.6$ | $28.3 \pm 0.0$ | $6.5 \pm 0.4$ | $8.9 \pm 0.1$ |
| | FRePo+VDSD | $22.0 \pm 1.0$ | $31.2 \pm 0.7$ | $8.6 \pm 0.1$ | $10.3 \pm 0.6$ |
| | IDTD | $22.5 \pm 0.1$ | $33.3 \pm 0.5$ | $9.5 \pm 0.3$ | $16.2 \pm 0.9$ |
| | Ours | $\mathbf{39.2 \pm 0.7}$ | $\mathbf{54.8 \pm 0.5}$ | $\mathbf{13.9 \pm 0.9}$ | $\mathbf{20.2 \pm 0.4}$ |

## 4.1 DATASETS

In this study, we conduct experiments on several widely-used video benchmark datasets, including UCF101(Soomro et al., 2012), HMDB51(Kuehne et al., 2011), Kinetics-400(Carreira & Zisserman, 2017), and Something-Something V2(Goyal et al., 2017). To ensure fair comparison with previous work (Wang et al., 2024; Zhao et al., 2024), we adopt the light-weight version of UCF101, namely MiniUCF.

UCF101 is a widely used benchmark dataset for human action recognition, containing 13,320 unconstrained video clips from 101 action categories. HMDB51 consists of 6,766 video clips spanning 51 human action classes. Kinetics-400 is a large-scale video dataset introduced by DeepMind, comprising around 240,000 training videos and 400 diverse action categories. Something-Something V2 focuses on fine-grained temporal reasoning and object interactions, with 220,847 video clips across 174 action categories.

## 4.2 IMPLEMENTATION DETAILS

To ensure fair comparison, all experimental settings are aligned with those used in VDSD (Wang et al., 2024) and IDTD (Zhao et al., 2024). For the MiniUCF and HMDB51 datasets, we follow the dataset splits provided by VDSD, and each video is sampled to 16 frames with a resolution of 112×112. For the Kinetics-400 and SSv2 datasets, each video is sampled to 8 frames with a resolution of 64×64. The four datasets are grouped into two categories for experimentation: MiniUCF and HMDB51 are treated as light-weight datasets, and we report top-1 accuracy; Kinetics-400 and SSv2 are considered large-scale datasets, and we report top-5 accuracy. For the proxy model, we adopt MiniC3D(Wang et al., 2024), a light-weight version of the C3D(Tran et al., 2015) model obtained through architectural simplifications. We will provide a detailed description of the model architecture in Appendix H.1.

## 4.3 MAIN RESULTS

We present the experimental results on light-weight and large-scale datasets in Table 1 and Table 2, respectively, demonstrating the superiority of our method across various datasets and IPC (instance per class) settings. The methods evaluated include coreset selection methods, image dataset distillation methods, existing video dataset distillation methods and our method.

**Light-weight Datasets.** On the MiniUCF and HMDB51 datasets, our method achieves significantly better performance compared to existing approaches. Specifically, under the IPC=5 setting on MiniUCF, our method attains an accuracy of 54.8%, yielding a 21.5% improvement over previous methods. On HMDB51 with IPC=5, our method achieves 20.2% accuracy, outperforming existing methods by 4%. Although the performance gain on HMDB51 appears smaller compared to MiniUCF,

Table 2: Comparison of top-5 accuracy with existing methods on large-scale datasets. $^\dagger$ denotes the top-5 accuracy of our teacher models trained on the full dataset is lower than the baselines due to different hyper-parameters settings.

| Dataset | Kinetics-400 | | SSv2 | |
|---|---|---|---|---|
| IPC | 1 | 5 | 1 | 5 |
| Random | $3.0 \pm 0.1$ | $5.6 \pm 0.0$ | $3.3 \pm 0.1$ | $3.9 \pm 0.1$ |
| DM | $6.3 \pm 0.0$ | $9.1 \pm 0.9$ | $3.6 \pm 0.0$ | $4.1 \pm 0.0$ |
| MTT | $3.8 \pm 0.2$ | $9.1 \pm 0.3$ | $3.9 \pm 0.1$ | $6.3 \pm 0.3$ |
| DM+VDSD | $6.3 \pm 0.2$ | $7.0 \pm 0.1$ | $4.0 \pm 0.1$ | $3.8 \pm 0.1$ |
| MTT+VDSD | $6.3 \pm 0.1$ | $11.5 \pm 0.5$ | $5.5 \pm 0.1$ | $8.3 \pm 0.1$ |
| IDTD | $6.1 \pm 0.1$ | $12.1 \pm 0.2$ | $3.9 \pm 0.1$ | $9.5 \pm 0.3$ |
| Ours | $\mathbf{6.5 \pm 0.2}^\dagger$ | $\mathbf{13.4 \pm 0.3}^\dagger$ | $\mathbf{10.4 \pm 0.1}$ | $\mathbf{15.2 \pm 0.2}$ |

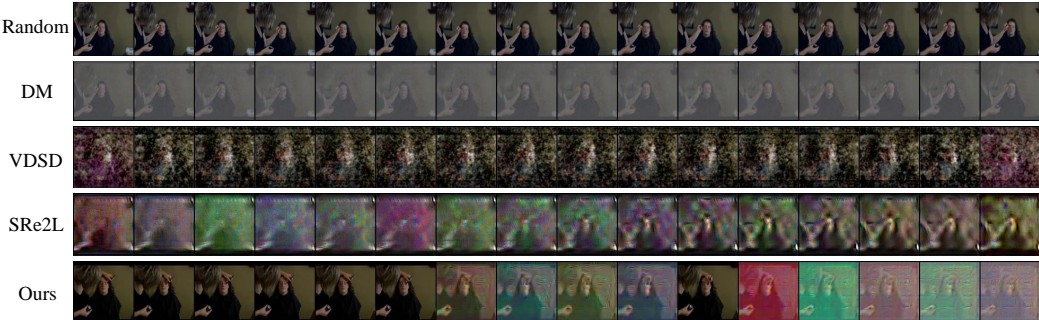

Figure 3: Visualization results on MiniUCF under IPC=1 with varying numbers of frames.

it is important to note that the full dataset accuracy on HMDB51 is only 28.6%. The accuracy achieved by our distilled dataset is already approaching that of the full dataset, demonstrating the effectiveness of our method. Coreset selection and image dataset distillation methods generally exhibit inferior performance, highlighting the limitations of image-based algorithms in effectively leveraging temporal information inherent in video data. These methods, when directly transferred to the video domain, can only achieve suboptimal results.

Existing video distillation algorithms address this issue by incorporating partial temporal cues, leading to moderate performance gains; however, their improvements remain limited, as their effectiveness still primarily relies on spatial information, with minimal exploitation of temporal dynamics. Our proposed distillation algorithm achieves consistent performance improvements across all experimental settings, validating the effectiveness of the proposed temporal saliency-guided filter.

**Large-scale Datasets.** Under the IPC=5 setting on the SSv2 dataset, our method achieves an accuracy of 15.2%, resulting in a 5.7% performance gain over existing approaches. On Kinetics-400, our method also surpasses the baseline, even though the model used achieves only 22.4% accuracy when trained on the full dataset. With an IPC=5 setting, the compression ratio on both datasets falls below 1%, making the distillation task significantly more challenging than on light-weight datasets. Consequently, the overall performance of existing methods remains relatively low, with a substantial gap compared to models trained on the full datasets. This highlights the considerable room for improvement in video dataset distillation on large-scale datasets. Despite the increased difficulty, our method consistently outperforms existing approaches across all experimental settings.

## 4.4 VISUALIZATION

To visually demonstrate the effectiveness of our method in utilizing temporal information, we conduct a qualitative analysis of the distilled results under the IPC=1 setting on the MiniUCF dataset, with visualizations for each method presented in Figure 3. As shown in the results, the distilled samples generated by the DM method are visually similar to real images. In contrast, VDSD suffers from

distortion due to its reliance on interpolation techniques, and SRe$^2$L's matching strategy leads to homogenization across frames, resulting in the loss of temporal information. In comparison, our method enables dynamic optimization of individual frames, effectively preserving temporal dynamics while simultaneously compressing spatial information. For additional visualization results, please refer to the Appendix G.

### 4.5 ABLATION STUDY

**Effectiveness of Each Component.** To verify the effectiveness of each component in the distillation framework, we conduct ablation studies on the MiniUCF dataset under the IPC=5 setting. The detailed results are shown in Table 3, where TSGF$_O$ denotes the temporal saliency-guided filter, and TSGF$_A$ refers to the temporal saliency-aware video augmentation strategy. The baseline refers to the three-stage video distillation framework without the inclusion of these two components. Experimental results demonstrate that both components contribute significantly to performance improvements.

Table 3: Ablation study on individual components. TSGF$_O$ and TSGF$_A$ denote the video optimization and the video augmentation based on TSGF, respectively.

| Methods | Acc |
| --- | --- |
| baseline | $40.5 \pm 0.3$ |
| baseline + TSGF$_A$ | $46.9 \pm 0.2$ |
| baseline + TSGF$_O$ | $51.7 \pm 0.6$ |
| baseline + TSGF$_O$ +TSGF$_A$ | $\mathbf{54.8 \pm 0.5}$ |

**Cross Architecture Generalization.** We present the cross architecture generalization results in Table 4, where all the evaluated model architectures are introduced in (Wang et al., 2024). Although the synthetic dataset is generated based on a pre-trained model, which limits the performance gain when transferring across architectures, our method consistently outperforms existing approaches across all architectures. This demonstrates the strong generalization ability of our method in cross-architecture scenarios.

Table 4: Cross-architecture experiments on Mini-UCF with IPC=1. Our method achieve superior generalization ability.

| | Evaluation Model | | |
| --- | --- | --- | --- |
| | ConvNet3D | CNN+GRU | CNN+LSTM |
| Random | $9.9 \pm 0.8$ | $6.2 \pm 0.8$ | $6.5 \pm 0.3$ |
| DM | $15.3 \pm 1.1$ | $9.9 \pm 0.7$ | $9.2 \pm 0.3$ |
| MTT | $19.0 \pm 0.1$ | $8.4 \pm 0.5$ | $7.3 \pm 0.4$ |
| DM+VDSD | $17.5 \pm 0.1$ | $12.0 \pm 0.7$ | $10.3 \pm 0.2$ |
| MTT+VDSD | $23.3 \pm 0.6$ | $14.8 \pm 0.1$ | $13.4 \pm 0.2$ |
| Ours | $\mathbf{39.2 \pm 0.7}$ | $\mathbf{17.8 \pm 0.4}$ | $\mathbf{16.5 \pm 0.3}$ |

**Number of Frames.** Figure 4 presents the accuracy comparison between our method and the baseline under different numbers of frames. As shown, when the number of frames is small, the video data essentially degrades into image data, with minimal temporal information available. In this case, our method achieves relatively poor performance. However, as the number of frames increases and temporal information becomes more prominent, the performance of our method improves significantly, demonstrating its effectiveness in leveraging temporal cues in video data.

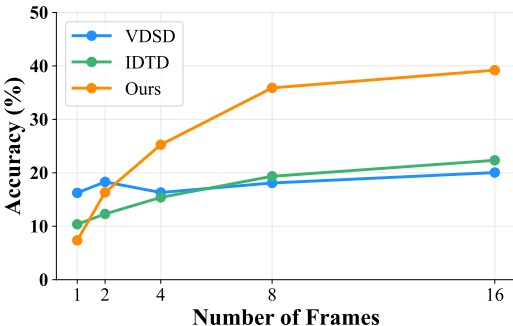

Figure 4: Experimental results on MiniUCF under IPC=1 with varying numbers of frames.

### 5 CONCLUSION

In this study, we propose a unified video distillation framework that enables efficient video distillation based on a Temporal Saliency-Guided Filter. We first train the model using a standard training process, and then perform temporally guided video optimization to simultaneously compress spatial and motion-related information during the distillation process. To further enhance video diversity, we adopt temporally guided video augmentation to augment the video data. Extensive experiments conducted on datasets of varying scales and domains, demonstrating that our method provides a novel and effective paradigm for video dataset distillation.

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

## TECHNICAL APPENDICES AND SUPPLEMENTARY MATERIAL

## A ABLATION STUDY ON INITIALIZATION

Table 5: Results under various initialization strategies

| Methods | Acc |
|---|---|
| random init | $36.8 \pm 0.3$ |
| real init | $54.8 \pm 0.5$ |

Table 6: Ablation study on data augmentation.

| Methods | Acc |
|---|---|
| image-based | $43.1 \pm 0.4$ |
| TSGF$_A$ | $54.8 \pm 0.5$ |

Table 7: Experimental results on static and dynamic group.

| IPC | Acc | S_Acc | D_Acc |
|---|---|---|---|
| 1 | $39.2 \pm 0.7$ | $35.5 \pm 1.0$ | $34.6 \pm 0.8$ |
| 5 | $54.8 \pm 0.5$ | $53.6 \pm 1.2$ | $58.5 \pm 0.7$ |
| 10 | $60.5 \pm 0.6$ | $57.1 \pm 0.9$ | $64.6 \pm 0.7$ |

To evaluate the impact of initialization on the distillation process, we conduct an ablation study by comparing different initialization strategies for the synthetic dataset. Specifically, we compare random noise initialization with initialization from real video frames. Our results indicate that real initialization significantly outperforms noise-based initialization. This suggests that temporal priors embedded in real videos provide a more informative starting point for learning temporal dynamics.

## B IPC ABLATION

We further examine the effect of instance per class (IPC) on the performance of the distilled dataset. Experiments are conducted on MiniUCF and HMDB51 with IPC values ranging from 1 to 20. As expected, the performance improves consistently with larger IPC values, highlighting the trade-off between data compactness and model accuracy.

Table 8: Ablation study on IPC.

| IPC | 1 | 5 | 10 | 20 |
|---|---|---|---|---|
| MiniUCF | $39.2 \pm 0.7$ | $54.8 \pm 0.5$ | $60.5 \pm 0.6$ | $62.8 \pm 0.2$ |
| HMDB51 | $13.9 \pm 0.9$ | $20.2 \pm 0.4$ | $22.5 \pm 0.9$ | $26.3 \pm 0.7$ |

## C DATA AUGMENTATION ABLATION

To understand the contribution of data augmentation in our framework, we conduct a comparative study between two settings: standard image-based augmentations and our proposed Temporally Guided Video Augmentation. The results clearly show that traditional image augmentations tend to disrupt temporal coherence, resulting in suboptimal performance. In contrast, our TSGF$_A$ significantly enhances the temporal diversity of synthetic videos while preserving motion continuity, leading to noticeable performance gains.

## D EXPERIMENTS ON STATIC AND DYNAMIC GROUP

Following the protocol established by VDSD(Wang et al., 2024), the MiniUCF dataset was partitioned into two subsets: the static group, consisting of categories characterized by minimal motion changes, and the dynamic group, comprising categories with significant temporal variations. We conducted separate distillation experiments on these two groups to investigate the effectiveness of our method under different temporal dynamics. Our results show that on the dynamic group, which requires capturing complex temporal dependencies, our method achieves outstanding performance. This demonstrates the superior capability of our framework in preserving and leveraging temporal dynamics during the distillation process. The distinction between static and dynamic groups highlights the importance of explicitly modeling temporal information in video dataset distillation. While static actions rely more on spatial cues, dynamic actions demand effective temporal modeling, which our method addresses through temporal saliency-guided optimization and augmentation.

# E    ABLATION STUDY ON GRADIENT DIRECTION

To investigate the effect of gradient direction, we conduct additional experiments in which the mask values are allowed to take negative values, enabling temporal saliency to modulate both the magnitude and direction of the gradients. We evaluate this setting on the MiniUCF dataset under various IPC configurations, as shown in Table 9. The results indicate that when the mask affects both the magnitude and direction of the gradients, the overall performance experiences a substantial degradation. This observation suggests that constraining the role of the mask to regulating gradient magnitude, rather than altering gradient direction, constitutes a more effective and robust strategy.

Table 9: Performance comparison of whether the mask affects the gradient direction.

| IPC | Magnitude-only | Both |
|-----|----------------|------|
| 1   | $39.2 \pm 0.2$ | $19.0 \pm 0.3$ |
| 5   | $54.8 \pm 0.5$ | $29.6 \pm 0.1$ |
| 10  | $60.5 \pm 0.3$ | $32.9 \pm 0.4$ |

# F    EXPERIMENTS ON DOWNSTREAM VIDEO TASKS

Existing baseline methods are primarily designed for dataset distillation in video classification tasks, and we follow the same setting to ensure fair comparisons. However, the proposed method offers a plug-and-play solution for video dataset distillation, as it does not rely on any task-specific supervision. This generality enables the method to be seamlessly extended and applied to other video understanding tasks beyond classification. Here we present some results by applying our method on Temporal Action Segmentation (TAS) dataset 50Salads (Stein & McKenna, 2013), as shown in Table 10. The results demonstrate that our method attains consistently strong performance on the TAS task.

Table 10: Performance on the Temporal Action Segmentation task.

|          | Acc  | Edit | F1@$10, 25, 50$ |
|----------|------|------|-----------------|
| Mean     | 69.0 | 42.7 | 50.0/46.1/37.4  |
| Coreset  | 61.7 | 43.3 | 49.9/46.3/35.4  |
| Ours     | **73.2** | **56.3** | **64.2/61.8/51.7** |

# G    VISUALIZATION

## G.1    OPTICAL FLOWS

To qualitatively analyze how our method preserves temporal dynamics, we visualize the optical flow of distilled videos in Figure 5. The optical flow maps clearly illustrate that our approach preserves smooth and consistent motion patterns. This indicates that the temporal dynamics crucial for action recognition are effectively retained during the distillation process.

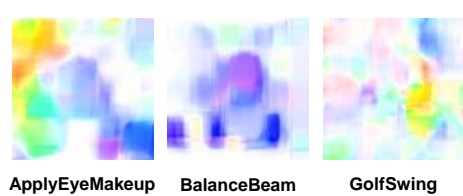

**ApplyEyeMakeup    BalanceBeam    GolfSwing**

Figure 5: Optical flows of MiniUCF.

## G.2    INTER-FRAME DIFFERENCES

In addition to optical flow, we also visualize frame differences to highlight temporal changes across frames. As shown in Figure 6, our method maintains coherent motion patterns and smooth temporal transitions. This further validates the effectiveness of our temporal saliency-guided framework in maintaining critical dynamic information necessary for accurate video understanding.

# H    IMPLEMENTATION DETAILS

## H.1    MODEL ARCHITECTURE

The specific architecture of MiniC3D is illustrated in Figure 7. For the convolutional layers, Conv3D 1 through Conv3D 3 adopt kernels of size 3×7×7 with a stride of 1×2×2 and padding of

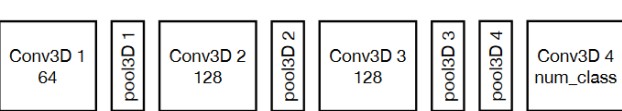

Figure 7: The architecture of MiniC3D.(Wang et al., 2024)

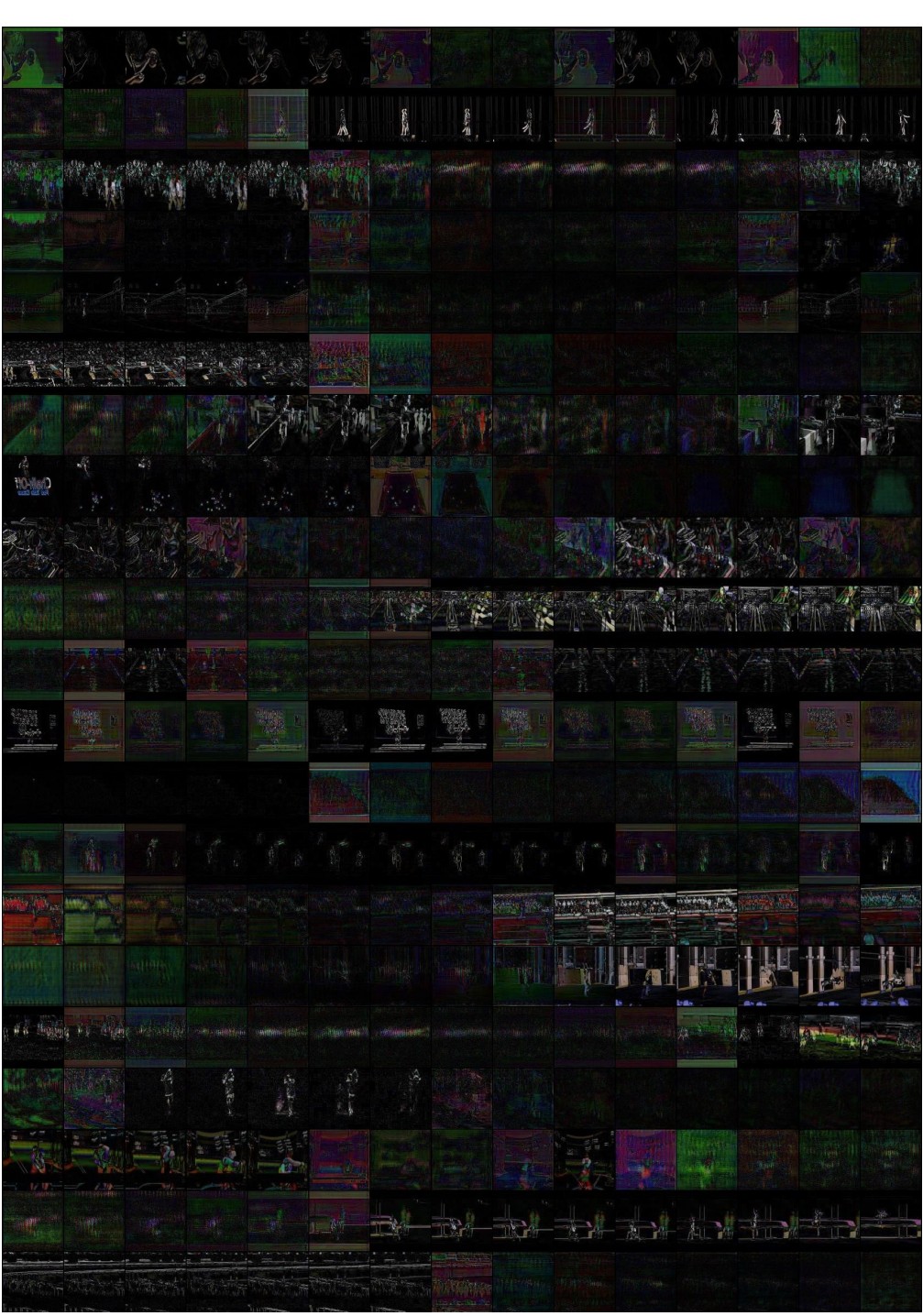

Figure 6: Inter-frame difference visualization on MiniUCF.

1×3×3. Conv3D 4, which serves as
the classification layer, employs a kernel of size 1×1×1 with a stride of 1×1×1. Regarding the pooling
layers, Pool3D 1 applies max pooling with a 1×2×2 kernel, Pool3D 2 and Pool3D 3 adopt max
pooling with a 2×2×2 kernel, while Pool3D 4 uses average pooling.

## H.2 HYPERPARAMETERS

We provide a detailed summary of the parameters used in
our experiments in Table 11, where lr denotes the learning
rate, and r_bn refers to the coefficient of the regularization
loss. Parameters not explicitly stated are assumed to follow
the default values specified in our code.

## H.3 COMPUTATIONAL RESOURCES

All experiments were performed on a server equipped
with eight NVIDIA RTX 3090 GPUs, each with 24 GB of
memory. Compared to training on the full-scale original
datasets, our video dataset distillation method significantly
reduces both training time and computational resource
demands.

| Dataset | IPC | lr | r_bn |
|---|---|---|---|
| MiniUCF | 1 | 0.25 | 0.001 |
|  | 5 | 0.25 | 0.005 |
| HMDB51 | 1 | 0.25 | 0.001 |
|  | 5 | 0.25 | 0.005 |
| Kinetics-400 | 1 | 0.3 | 0.01 |
|  | 5 | 0.3 | 0.01 |
| SSv2 | 1 | 0.3 | 0.01 |
|  | 5 | 0.3 | 0.01 |

Table 11: Hyperparameters for different datasets.

## I BROADER IMPACT

Our proposed video dataset distillation framework holds significant potential to advance research
and applications in video understanding by substantially reducing the computational cost associated
with training deep video models. This can democratize access to large-scale video analysis tech-
niques, enabling wider adoption across academia and industry, especially for groups with limited
computational resources. By facilitating faster and more efficient model training, our method may
accelerate developments in areas such as video surveillance, autonomous driving, human-computer
interaction, and content recommendation systems, ultimately benefiting society through improved
safety, convenience, and personalized experiences. Importantly, the distilled synthetic datasets gener-
ated by our approach inherently protect privacy by reducing reliance on storing and sharing large
volumes of original video data, which often contain sensitive or personally identifiable information.
From an ethical perspective, our work does not introduce new risks for malicious use, as it focuses
on improving data efficiency rather than altering or generating misleading content. Nonetheless,
as with any technology related to video analysis, responsible deployment should consider privacy,
consent, and potential biases in training data. Overall, we believe our method contributes positively
to sustainable and ethical AI development in the video domain.

