# OpenReview forum: "Temporal Saliency-Guided Distillation: A Scalable Framework for Distilling Video Datasets"
_ICLR.cc/2026/Conference — Submitted to ICLR 2026_

### Official Review · Reviewer_aNEz · 2025-10-29

[review text omitted: it was posted to a different submission]

---

> ### Author Response · Authors · 2025-11-30
>
> Thank you very much for your comments. However, we are afraid we could not locate the corresponding part of the issue you mentioned in our paper. We would greatly appreciate it if you could kindly provide further clarification.

---

### Official Review · Reviewer_8CWt · 2025-10-30

**Soundness:** 3
**Presentation:** 2
**Contribution:** 2
**Rating:** 4
**Confidence:** 4

**Summary:**

This paper presents a novel and efficient video dataset distillation method that leverages temporal saliency to preserve motion dynamics while compressing video data. The proposed framework achieves superior performance across multiple benchmarks with lower computational cost compared to existing methods.

**Strengths:**

1. The TSGF mechanism is an innovative and lightweight way to capture temporal importance without relying on heavy optical flow or 3D convolutions.
2. The model design is computationally efficient, adopting uni-level optimization design instead of complex bi-level optimization, which substantially reduces memory footprint and training cost, making it suitable for large-scale video data distillation.
3. The method achieves state-of-the-art results on multiple standard benchmarks (UCF101, HMDB51, Kinetics-400, SSv2), maintaining strong performance even under extreme compression ratios (e.g., IPC = 1).
4. The experimental evaluation is extensive, covering diverse datasets, compression ratios, architectures, and ablation studies, providing convincing empirical support for the proposed framework.

**Weaknesses:**

1. The paper lacks a formal theoretical definition and justification of “temporal saliency,” and the TSGF design appears heuristic, without mathematical modeling or convergence analysis.
2. Using raw inter-frame differences may not capture complex motion semantics and may fail in scenes with camera motion or background clutter.
3. Although the method claims potential applicability to other video tasks, only preliminary results are presented for temporal action segmentation, without validation on detection, tracking, or video generation tasks.
4. The presentation quality could be improved; for instance, the font sizes in Figure 2 appear inconsistent, and Tables 3 and 4 use mismatched formatting, which slightly affects readability.

**Questions:**

1. Further clarification on the rationale for using frame differencing as the saliency measure would be valuable, possibly supported by additional theoretical analysis or empirical ablation to justify its necessity and effectiveness.
2. It remains unclear whether TSGF can robustly handle diverse motion types—including rigid, non-rigid, and camera-induced motion—and whether there exist failure cases for specific video categories.
3. The single-stage optimization strategy might be prone to local minima; theoretical or experimental analysis of its convergence and stability, especially on complex video datasets, would strengthen the paper.
4. A deeper discussion comparing the proposed temporal modeling with existing approaches (e.g., optical flow, 3D convolutional features, or recurrent architectures) would help highlight the unique advantages of the proposed method.

---

> ### Author Response · Authors · 2025-11-30
>
> >**Weakness 1 & Question 1: Effectiveness of frame differencing.**
>
> In Table 3, we conducted empirical ablation experiments, and we observe that both TSGF_A and TSGF_O, which incorporate frame differencing, significantly improve performance compared with the baseline.
>
> >**Weakness 2 & Question 2: Performance of frame differencing in complex motion scenarios.**
>
> Frame differencing is used to adjust the optimization magnitude because incorporating optical flow or other motion cues would introduce prohibitive spatiotemporal overhead. As shown in Figure 1, dataset distillation for videos corresponds to distillation on an image dataset with a very large IPC, resulting in extremely high computational cost. Therefore, it is essential to adopt an optimization strategy that is as efficient as possible.
>
> In terms of effectiveness, SSv2 is a motion-centric dataset, and our method achieves substantially better performance than other approaches on this dataset, as shown in Table 2. This demonstrates the effectiveness of our approach for action-dominant datasets.
>
> >**Weakness 3: Other tasks.**
>
> Our method provides a plug-and-play distillation mechanism, and the main task remains video classification. The experiments on other tasks are included only as simple demonstrations.
>
> >**Weakness 4: Figure and table formatting issues.**
>
> Thank you for the feedback. We will revise these aspects in the full version of the paper.
>
> >**Question 3: Convergence and stability of the single-stage optimization strategy.**
>
> In each distillation process, the synthetic dataset is randomly initialized. As shown in Table 2, our experiments on large-scale datasets report results over multiple random trials, and the accuracy remains within a stable range. This demonstrates the strong stability of our method.
>
> >**Question 4: Comparison with temporal modeling methods.**
>
> The temporal modeling methods mentioned here cannot be directly integrated into the distillation process, and they also introduce substantial spatiotemporal overhead. How to design a distillation framework that incorporates existing temporal modeling techniques remains an open research question. Our work presents the first attempt that explicitly focuses on modeling temporal information within video dataset distillation.

---

### Official Review · Reviewer_D4hu · 2025-11-01

**Soundness:** 3
**Presentation:** 2
**Contribution:** 2
**Rating:** 4
**Confidence:** 4

**Summary:**

The paper proposes a uni-level video dataset distillation framework
that directly optimizes synthetic videos against a fixed, pre-trained classifier while enhancing temporal fidelity
through a Temporal Saliency Guided Filter (TSGF).
The framework involves three key stages: 1. training a teacher model on real videos;
2. optimizing synthetic videos by aligning cross-entropy loss and batch normalization statistics with the teacher model;
and 3. applying a saliency-guided video augmentation that emphasizes motion-consistent regions.
Temporal saliency is computed from inter-frame differences with smoothing and used both to modulate gradient magnitude during optimization
and to adaptively select augmentation regions. Experiments on diverse video datasets demonstrate consistent improvements
over prior dataset distillation methods, with ablations showing the contribution of each module.

**Strengths:**

1. Conceptually clean and efficient framework:
The uni-level training scheme is straightforward yet effective,
avoiding iterative teacher-student feedback while maintaining strong performance.

2. Temporal Saliency Guided Filter (TSGF): The proposed TSGF provides a principled way to incorporate motion cues
without requiring optical flow or explicit temporal modeling, which enhances both interpretability and efficiency.


3. Comprehensive empirical validation: The framework achieves consistent gains across diverse datasets
and varying compression ratios.
Reported results show competitive scalability even under efficient distillation budgets.

4. Potential for broader application: The idea of saliency-weighted optimization could generalize
to other video understanding or dynamic scene compression tasks.

**Weaknesses:**

1. Insufficient positioning relative to decoupled dataset distillation methods.
The pipeline (Sec. 3.2, Eq. (4)–(5)) resembles decoupled optimization schemes such as SRe2L,
where a frozen teacher guides the synthetic data optimization.
The distinction between the proposed uni-level framework and existing decoupled methods is not clearly explained,
leaving the degree of conceptual novelty somewhat ambiguous.



2. Incomplete hyperparameter specification for the Temporal Saliency Guided Filter.
Critical parameters in Eq. (7) and Eq. (8), such as the smoothing window size $k$, weighting coefficients $\alpha_k$,
and the $\epsilon$ constant for stability, are not reported.
No sensitivity or ablation analysis is presented for these factors, which limits reproducibility and understanding of robustness.


3. Limited robustness and failure mode analysis.
While the paper claims the TSGF improves robustness under motion variation,
there is no quantitative test for scenarios with strong camera motion or jitter.
Failure analysis is limited to coarse class splits in Tab. 7 without qualitative inspection.

4. Restricted evaluation scope across model architectures.
The cross-model evaluation (Tab. 4) considers only small CNN-based backbones.
There is no validation on transformer-based or modern video architectures, which limits the generalizability claim.
Additionally, quantitative comparison with a modern decoupled baseline (e.g., SRe2L) is missing, with only a qualitative reference in Fig. 3.

**Questions:**

1. It would improve clarity to relocate Fig. 2 (framework overview) to the beginning of the Method section
and Alg. 1 (optimization steps) to its end for smoother narrative flow.

---

> ### Author Response · Authors · 2025-11-30
>
> >**Weakness 1: Novelty and effectiveness.**
>
> Existing decoupled methods are designed for images only; directly applying them to videos causes degradation across frames, leading to homogenization where the distilled video collapses into a stack of nearly identical images. To prevent this issue, we introduce TSGF, which adaptively controls the optimization of each frame so that the distilled videos can preserve video-specific temporal characteristics.
>
> In Table 3, the baseline refers to the direct adaptation of the decoupled distillation algorithm to videos. As shown, simply applying the decoupled method results in only 40.5% accuracy, while our method achieves 54.8%.
>
> >**Weakness 2: Hyperparameter explanation.**
>
> The weighting coefficients are generated using the Hanning window function. Constants such as the smoothing window size and other unspecified parameters will be clearly provided in our released code.
>
> We also include an sensitivity analysis on r_{bn} with IPC=5 on MiniUCF, as shown below:
> |$r_{bn}$|Acc|
> |-|-|
> |0.0005|53.7|
> |0.005|54.8|
> |0.05|50.6|
> |0.1|49.9|
>
> >**Weakness 3: Failure analysis and robustness under motion variation.**
>
> This work focuses on dataset distillation, whose goal is to compress the useful information of the full dataset into a limited number of synthetic samples. The distilled samples do not correspond to any specific original instance but instead represent the dataset’s overall distribution. Therefore, they are not suitable for analyzing a single specific scenario.
>
> Regarding robustness under motion variation: compared with the baseline, which primarily captures spatial information, our method is better at compressing motion-related semantics. This is supported by the SSv2 results in Table 2 and the dynamic-group experiments in Table 7.
>
> >**Weakness 4: Experiments on ViT architectures and comparison with decoupled methods.**
>
> For a fair comparison with the baseline, we mainly use CNN-based architectures. The comparison with SRe2L can be found in Weakness 1.
>
> >**Question 1: Image placement adjustment.**
>
> Thank you for the feedback. We will update the full version of the paper accordingly.

---

### Official Review · Reviewer_tVoB · 2025-11-07

**Soundness:** 4
**Presentation:** 4
**Contribution:** 3
**Rating:** 4
**Confidence:** 3

**Summary:**

This method proposes a uni-distillation framework for video datasets, introducing a Temporal-Spatial Gating Function (TSGF) for automatic frame attention and data augmentation during post-training. The approach achieves advanced results on various benchmarks, demonstrating robustness to dataset scale and motion strength.

**Strengths:**

1. Proposed method is effective at pruning redundant frames, making the distillation process significantly faster and more tractable for massive datasets and achieves new state-of-the-art accuracy on multiple benchmarks.
2. The TSGF is a well-motivated addition that explicitly addresses the challenge of preserving temporal dynamics, a critical weakness in previous video distillation methods.

**Weaknesses:**

1. Dataset distillation maintains performance with significantly smaller synthetic datasets; however, the baseline and its variants appear to underperform (e.g., 22.4% on K400). Since smaller models tend to overfit small-scale datasets, they may not adequately demonstrate the generalization capability of distilled datasets. While Table 4 shows that this method benefits larger models, could the authors further validate this finding using video models of standard scale rather than those with only a few layers?
2. The performance improvements differ between background-based datasets (K400) and motion-based datasets (SSv2), yet this paper lacks analysis of these differences. Additionally, marking the average number of samples per class would facilitate better understanding of the performance variations.
3. What are the sizes of the distilled dataset and the full dataset for MiniUCF, and what is the accuracy achieved on the full dataset?

**Questions:**

Please refer to Weakness

---

> ### Author Response · Authors · 2025-11-30
>
> >**Weakness 1: Experiments on complex models.**
>
> We have added experiments using the I3D [1] model on MiniUCF. The detailed results are shown in Table 1.
>
> |IPC|ConvNet3D|I3D|
> |-|-|-|
> |1|39.2|23.7|
> |5|54.8|55.5|
> |10|60.5|71.8|
>
> >**Weakness 2: Explanation on performance difference and dataset sizes.**
>
> TSGF effectively preserves temporal information during the distillation process. Therefore, on motion-based datasets such as SSv2, TSGF achieves significantly better performance than the baseline. In contrast, background-based datasets focus primarily on spatial cues, so the performance gain of TSGF over the baseline is less pronounced.
>
> The average number of samples per class for each dataset is as follows: MiniUCF: 132, HMDB51: 134, Kinetics400: 636, and SSv2: 1269.
>
> >**Weakness 3: Details of the full dataset.**
>
> MiniUCF contains approximately 5,000 videos, and the accuracy achieved on the full dataset is 57.22%.
>
> [1] Joao Carreira and Andrew Zisserman. Quo vadis, action recognition? a new model and the kinetics dataset. In proceedings of the IEEE Conference on Computer Vision and Pattern Recognition, pp. 6299–6308, 2017.

---

### Meta-Review · Area_Chair_wFL9 · 2026-01-10

**Summary:**

Apparently, Reviewer aNEz does not provide a review that matches this paper, so we will ignore its review. One concern is the insufficient technical novelty, as the difference between the proposed method and existing decoupled methods is unclear or significant, although the existing decoupled methods are designed for image datasets. There are some unclear details from reviewers for clarification, such as the analysis of performance differences between background-based and motion-based datasets, and critical hyperparameters for TSGF. There is also concern about the generalization ability of the proposed approach because the evaluation is limited to small CNN architectures, missing transformer-based models and modern architectures.

**Reviewer Concerns:**

Some clarification questions are addressed by the rebuttal. However, I think the technical novelty issue is not well addressed. Eq. (4)–(5) in the paper are not tailored for videos, which look similar to existing decoupled methods, as pointed out by reviewers. There is no evaluation on transformer architecture, which casts doubt on whether the proposed method can work well for other modern models. Thus, the AC does not recommend acceptance to ICLR.

**Reviewer Scores:**

We should ignore the scores of  Reviewer aNEz. The final scores should be 4,4,4. I think the scores will be unchanged if there is active discussion.

---

### Decision · Program_Chairs · 2026-01-26

Reject